# The Role of Cytochrome P450 4C1 and Carbonic Anhydrase 3 in Response to Temperature Stress in *Bemisia tabaci*

**DOI:** 10.3390/insects12121071

**Published:** 2021-11-30

**Authors:** Xiaona Shen, Wanxue Liu, Fanghao Wan, Zhichuang Lv, Jianying Guo

**Affiliations:** 1State Key Laboratory for Biology of Plant Diseases and Insect Pests, Institute of Plant Protection, Chinese Academy of Agricultural Sciences, Beijing 100193, China; 82101181813@caas.cn (X.S.); liuwanxue@caas.cn (W.L.); wanfanghao@caas.cn (F.W.); 2Agricultural Genome Institute at Shenzhen, Chinese Academy of Agricultural Sciences, Shenzhen 518120, China

**Keywords:** cytochrome P450, carbonic anhydrase, chromatin accessibility, invasive species, temperature adaptation

## Abstract

**Simple Summary:**

Temperature is an important factor affecting all physiological activities of ectotherms. Invasive whiteflies can quickly adapt to new environments probably regulated by epigenetics. The results of a chromatin openness test also showed that the position of the chromatin opening of *Bemisia tabaci* undergoes significant change under different temperature stresses. However, the specific regulatory factors in this process have not yet been verified. In this study, we verified two key factors, cytochrome P450 4C1 and carbonic anhydrase 3, regulated by chromatin accessibility. Our findings can provide a potential mechanism for responses to temperature stress and a direction for other behavioral activities of insects, as well as in proposing strategies for controlling invasive populations of whitefly.

**Abstract:**

The position of the chromatin opening of *Bemisia tabaci* undergoes significant changes under different temperature stresses, and numerous regulatory factors have been found. In this study, we verified two key factors, cytochrome P450 4C1 and carbonic anhydrase 3. The results showed that invasive whiteflies had a significantly lower heat resistance after silencing *BtCYP 4C1* and *BtCar3*. In addition, whiteflies had a higher cold tolerance after silencing *BtCYP 4C1*. These results indicate that *BtCYP 4C1* and *BtCar3* are key regulators in the temperature adaptation of *B. tabaci*. Moreover, they may be key factors in influencing the geographical distribution and dispersal of *B. tabaci* as an invasive species in China.

## 1. Introduction

Insects are ectothermic animals. Therefore, temperature affects their physiological activities, including growth, development, reproduction and survival. In their natural habitats, it is also found that different species of insects have different temperature tolerances [1], which might be one of the factors leading to the wide distribution of some invasive insects, such as *Liriomyza* spp., *Hyphantria cunea* and *Bemisia tabaci* [2,3,4,5].

*Bemisia tabaci* Middle East-Asia Minor (MEAM) cryptic species, also referred to as the B biotype, is an invasive pest that causes damage to many crops through direct feeding, the deposition of honeydew and the transmission of plant viruses [6,7,8,9,10,11]. In the 1990s, with the strengthening of full-scale trade, *B. tabaci* MEAM cryptic species successfully invaded China with the help of flowers and other cash crops and is now widely distributed across most provinces [10]. Many studies have indicated that the temperature adaptation processes of *B. tabaci* are regulated at the molecular level [12,13,14,15,16]. In a temperature adaptability study of *B. tabaci*, it was found that the whitefly can quickly adapt to a new environment when entering a new habitat, and its temperature tolerance has produced corresponding transient genetic changes [17,18]. Epigenetic regulation is a way of regulating the phenotype of the organism without changing gene sequences in a short period of time. Several studies showed that temperature adaptability has likely allowed invasive *B. tabaci* haplotypes to expand to new environmental niches. These adaptation processes have been described as transient phenotypic changes, likely mediated by epigenetic regulation, where in the genetic sequence remains unchanged. These results support the hypothesis that the adaptability of *B. tabaci* to temperature stresses is mediated epigenetically. In such a specific chromatin accessibility pattern, regulatory regions were found to harbor thirty genes of interest, among which, cytochrome P450 4C1 (*BtCYP 4C1*) and carbonic anhydrase 3 (*BtCar3*) we selected based on the significance of the difference in their transcriptional expression levels and chromatin accessibility. They have also been found, in other insects, to play a role in the process of resistance [19,20,21,22,23]. Nevertheless, the role of *BtCYP 4C1* and *BtCar3* in the temperature stress process of *B. tabaci* has not been confirmed.

In order to verify the role of *BtCYP4C1* and *BtCar3* in the response of *B. tabaci* to temperature stress, the silencing of *BtCYP4C1* and *BtCar3* was carried out using an RNAi approach and whiteflies were fed on specific dsRNAs, followed by exposure to heat and cold stress regimes. The timing of recovery from temperature shocks was recorded as a measurement of the effect of the dsRNAs on whiteflies. These results showed that the two genes played a role in *B. tabaci* resistance to adverse to environmental temperature. It is further confirmed that *BtCYP 4C1* and *BtCar3* are the key regulatory factors involved in the temperature adaptation regulated by chromatin accessibility, providing a new perspective on the molecular mechanisms regulating temperature tolerance.

## 2. Materials and Methods

### 2.1. Experimental Insects

*B. tabaci* cryptic species MEAM, were reared on cotton plants in greenhouse conditions. Insects were maintained in cages in an insectary at 24–26 °C under 50–60% relative humidity with a 14:10 h light: dark cycle. Twenty *B*. *tabaci* adults were randomly selected to identify the species using cytochrome oxidase subunit 1 gene to ensure that the whiteflies are MEAM cryptic species once a month.

### 2.2. RNA Extraction and cDNA Synthesis

Total RNA was isolated from approximately 200 MEAM adults using TRIzol reagent (Invitrogen, Carlsbad, CA, USA) following the manufacturer’s protocol. RNA was quantified using a NanoPhotometer TM P330 instrument (Implen, Munich, Germany), and the A260/A280 ratio was typically above 2.0. The RNA quality was also evaluated via 1% agarose gel electrophoresis. The RNA was treated with DNase to avoid DNA contamination. Reverse transcription was performed using 2.0 μg of each RNA sample in a 20.0-μL reaction with an oligo (dT)18 primer (Transgen, Beijing, China) according to the instructions provided with a Super Script First-Strand Synthesis System (Transgen).

### 2.3. Sequence Analysis

The coding region sequence of the target genes were found from the transcriptome database(unpublished). Sequence alignment and identity analyses were performed using DNAMAN (version 5.0; LynnonBioSoft, Quebec, QC, Canada). Molecular weights and pIs were calculated using ExPASy (http://web.expasy.org/protparam/, accessed on 1 September 2021). Conserved functional domains of the deduced protein sequences of the two genes were identified using SMART software (http://smart.embl-heidelberg.de/, accessed on 1 September 2021). Signal peptides and cross membrane domains were predicted using SignalP 5.0 Server (http://www.cbs.dtu.dk/services/SignalP/, accessed on 1 September 2021) and TMHMM Server v. 2.0 (http://www.cbs.dtu.dk/services/TMHMM/, accessed on 1 September 2021). Multiple protein sequences were aligned using DNAMAN and implemented in the MEGA 7 software package to evaluate the molecular evolutionary relationships. The phylogenetic tree was constructed with the neighbor joining method, using MAGE 5. Bootstrap majority consensus values for 1000 replicates are indicated at each branch point (%).

### 2.4. Quantitative Real-Time PCR Analysis of Relative Expression Levels

The effect of gene silencing on temperature tolerance in whiteflies fed dsRNA were assessed. The relative mRNA expression level was analyzed by qPCR. The primer sequences used are listed in Table 1. The reactions were performed using an ABI 7500 Real-time PCR system (Applied Biosystems, Waltham, MA, USA). All amplifications were confirmed by sequencing, and the specificity of qRT-PCR reactions was estimated by melting curve analysis. PCR assays were prepared to a final volume of 20.0 μL with 1.0 µL of the cDNA template, 10.0 µL of 2× TransStart TM Green qPCR SuperMix, 200 µM of each gene-specific primer (Table 1), and 0.4 µL of passive reference dye. A thermocycler was programmed with the following cycling conditions: (1) 94 °C for 1 min, followed by (2) 40 cycles of 95 °C for 15 s, 61 °C for 30 s and 72 °C for 30 s. A control without the cDNA template was included in all batches. *EF1-α* was used as the reference gene because it is constitutively expressed under various temperature stress conditions [24].

Amplification efficiency was validated by constructing a standard curve using six serial dilutions of cDNA. The relative quantification of mRNA expression was calculated using the mathematical model [25], which simplifies to 2^−ΔΔCT^ as follows: (ΔΔCT = (Ct target − Ct reference) treatment − (Ct target − Ct reference) control). The relative mRNA expression level was defined as the fold change compared to the amounts of *EF1-α*. Each sample was assessed in triplicate (technical replicates).

### 2.5. RNA Interference and Phenotype Observation

To synthesize dsRNA, two fragment templates of *BtCYP*
*4C1* (302 bp) and *BtCar3* (331 bp) were amplified by PCR using cDNAs cloned previously as templates with forward and reverse primers containing the T7 primer sequence (Table 1) at the 5′ ends. The sequences were verified by sequencing (Invitrogen). In-vitro dsRNA synthesis was performed using the MEGAscript T7 High Yield Transcription Kit (Ambion, Austin, TX, USA). dsRNA was quantified using a NanoPhotometer TM P330 instrument (Implen, Munich, Germany). The concentration of the synthesized dsRNA is around 9000 ng/μL.To deliver the dsRNA into the body of whiteflies, we fed individuals diet containing dsRNA diluted to 0.4 μg/μL in a 10% *w*/*v* RNase-free sucrose solution using the Parafilm clip nutrient solution method [26]. The Parafilm was pre-treated with 0.1% DEPC solution to remove any RNase, and RNase-free water was used to clean DEPC from the Parafilm. The control whiteflies were fed with an enhanced green fluorescent protein (EGFP)-specific dsRNA.

Whitefly adults (300 individuals) were collected randomly and placed in separate glass tubes (3 cm in diameter × 8 cm in height) for the RNA interference experiment. A tube was then wrapped with black plastic paper, leaving the Parafilm-enclosed end exposed to light to encourage the adults to move towards to the direction of light and consume the sucrose solution supplied. After feeding for 4 h, approximately 200 samples were immediately frozen in liquid nitrogen for detecting relative expression levels of genes by qPCR. Then the remaining whiteflies were exposed to high (45 ± 0.2 °C) and low temperature (−5 ± 0.2 °C) stress for 10 min in a water bath, 40 adults respectively, after removing the damaged whiteflies. The knock down and recovery times were recorded immediately. Each treatment had four biological replicates. The temperatures of 45 °C and −5 °C were selected based on preliminary experiments showing that these temperatures were threshold points for whitefly temperature tolerance.

### 2.6. Statistical Analysis

Statistical analyses were performed using GraphPad Prism 8.0 (GraphPad Software, San Diego, CA, USA) or Microsoft Excel (Microsoft, Redmond, WA, USA). All percentage data were log-transformed to ensure that they were normally distributed. Target gene mRNA expression data and the knock down and recovery times after feeding with the dsRNA mixture were analyzed used Student’s *t*-test. The data were presented as the mean ± SEM. Differences were considered statistically significant when *p* ≤ 0.05.

## 3. Results

### 3.1. Sequence and Characterization of BtCYP 4C1

The cytochrome P450 4C1 gene *BtCYP 4C1* (GenBank accession number: OK239710) in the MEAM cryptic species has an open reading frame of 1506 bp, encoding 501 amino acid residues with a predicted molecular weight of 57.18 kDa and pI of 8.38. The deduced amino acid sequence had a transmembrane structure (2–21) and contained a 465 aa Pfam domain (34–498) in the cytochrome P450 4C1 gene (Figure 1A). Furthermore, the cytochrome P450 4C1 gene is enriched in the oxidation-reduction process (GO:0055114). Similarities to other species were between 35% and 40% (*Nilaparvata lugens*: XP_039277931.1, 36.69%; *Halyomorpha halys*: XP_014272659.1, 35.43%; and *Thrips palmi*: XP_034246573.1, 37.74%). Although the similarity is low, we found that the amino acid residues at some positions are highly conserved through multiple sequence alignment of cytochrome P450 4C1 from different species (Figure 1B). The result showed the conservation of the gene function.

To examine the phylogenetic relationships among *BtCYP 4C1* and its homologues in other insects, a phylogenetic tree was constructed based on the deduced amino acid sequences from 17 species, which included 7 orders. As shown in Figure 1C, not all the insects of Hemiptera clustered together on a large branch. This result implies that CYP 4C1 is likely to have unique functions in different species.

### 3.2. Sequence and Characterization of BtCar3

The carbonic anhydrase 3 gene *BtCar3* in the MEAM (GenBank accession number: OK239710) cryptic species has an open reading frame of 903 bp, encoding 300 amino acid residues, with a predicted molecular weight of 34.77 kDa and pI of 5.32. The deduced amino acid sequence has a signal peptide (1–21) and contained a 261-aa carbonic anhydrase domain (33–293) in the carbonic anhydrase 3 gene (Figure 2A). It is a eukaryotic-type carbonic anhydrase belonging to cytosolic CAs. It is worth mentioning that carbonic anhydrase of many species, (including *Aphis gossypii* and *Halyomorpha halys* from Hemiptera and humans) had no signal peptide predicted using SignalP 5.0 Server, suggesting the unique role of carbonic anhydrases on whitefly.

Phylogenetic analysis was performed using the coding region of *BtCar3* to determine its evolutionary pattern among 19 species, which included six orders. As shown in Figure 2B, most of gene Car1 and Car2 could be clustered together; that is, the genetic distance between them is relatively close. By contrast, Car3 could not cluster well with them. Considering that carbonic anhydrase genes are widely studied in humans, we conducted a separate phylogenetic tree analysis of the three carbonic anhydrase genes from humans and *B. tabaci*. Surprisingly, the Car3 gene of *B. tabaci* is more likely to merge with three human carbonic anhydrase genes (Figure 2C). Therefore, we performed a multiple alignment analysis based on the deduced amino acid sequences of three carbonic anhydrase genes from *B. tabaci* and humans (Figure 2D). The results showed that the amino acid residues, at some positions of the Car3 gene sequence in *B. tabaci,* are the same as the human carbonic anhydrase genes. This result further hints at the specificity of carbonic anhydrase 3 in driving function in the whitefly.

### 3.3. mRNA Expression in B. tabaci after dsRNA Feeding

Real-time PCR was used to detect the relative expression levels of mRNA after the dsRNA feeding of whiteflies. Melting curve analysis indicated that the primers used were specific for the *BtCYP 4C1* and *BtCar3* genes. Compared with expression in the control groups (fed with dsRNA of EGFP), the expression of mRNA in the respective treatment groups of each gene decreased significantly (Figure 3). The silencing efficiency of ds*BtCYP 4C1* was 86.70%, and the silencing efficiency of ds*BtCar3* was 79.37%. The results show that RNAi reduced the expression of target genes effectively in B. tabaci and the results of the phenotypic observation are robust.

### 3.4. Effects of dsBtCYP 4C1-Feeding on Temperature Resistance

The specific function of *BtCYP 4C1* in temperature resistance in *B.tabaci* was investigated by comparing the means of heat (45 °C) knockdown time (T_KD_) and chill-coma (−5 °C) recovery time (T_RC_) after the silencing of the targeted genes between the treatment and control groups. T_KD_ and T_RC_ values were all significantly lower in whiteflies fed ds*BtCYP 4C1* than that of ds*EGFP* (Figure 4), indicating significantly lower heat resistance and higher cold resistance (Figure 4).

### 3.5. Effects of dsBtCar3-Feeding on Temperature Resistance

Compared with control groups, the T_KD_ values of the ds*BtCar3* treatment were significantly lower (Figure 4), while the T_RC_ values had not significantly changed (Figure 4) in *B. tabaci*. Data from phenotype observations show that the knockdown of *BtCar3* had a deleterious effect on heat resistance in whitefly adults.

## 4. Discussion

Compared with native species, invasive species can quickly adapt to an environment after entering new habitats, while gene mutations need more time to accumulate. Therefore, epigenetic mechanisms that can change phenotype without changing DNA sequence are very likely to play a role in the rapid adaptation process. Therefore, it is necessary to explore the effect of epigenetic mechanisms in rapid adaptation, which is helpful in controlling invasion by invasive species.

Transcript levels of cytochrome P450 genes in pine wood nematodes (PWNs) were elevated at low temperature, and the knockdown of these genes decreased the survival rates of PWNs under low temperature [19]. A study on low-temperature stress for the rose showed that two cytochrome P450 genes were up-regulated. The results implied that cytochrome P450 genes are involved in the process of the rose’s response to low temperatures [20]. Evidence suggests that temperature preference behavior, in *Drosophila melanogaster*, is regulated by the cAMP-dependent protein kinase (PKA)-Cytochrome P450 signaling pathway [21]. The above studies have all showed that cytochrome P450 plays a certain role in the temperature response process. In this study, it is confirmed that silencing cytochrome P450 4C1 genes affects the high temperature tolerance of the whitefly. This result implies that the process of CYP4C1 in regulating temperature tolerance is a rapid response process. In other words, it can regulate organisms to adapt to different temperatures quickly and enhance their viabilities.

Carbonic anhydrases (CAs) are metabolic enzymes that regulate the physiological equilibrium in a variety of organisms and play a vital role in mammals. In several insects, carbonic anhydrase has been found to be directly related to pesticide resistance [23,24]. Beta carbonic anhydrases were regarded as novel targets for pesticides and anti-parasitic agents in agriculture and livestock husbandry [27]. All of these results indicate that carbonic anhydrases perform a very important function in insects. In the *B. tabaci* genome, a total of four types of carbonic anhydrase were found; and, in this study, high-temperature tolerance decreased after interfering with the expression of *BtCar3* in *B. tabaci*. This result implies that *BtCar3* is a key regulator in the process of the epigenetic regulation of temperature adaptation in *B. tabaci*.

In this study, we verified two regulatory factors of the *B. tabaci* quick responding to temperature stress. Although additional studies detailing the pathways of *BtCYP 4C1* and BtCar3 are needed to further elucidate the epigenetic mechanisms of response to stress. The results of this study can still provide a potential mechanism of such response to temperature stress and a direction for studying other behavioral activities of insects, as well as in proposing strategies to control invasive populations of whitefly.

## Figures and Tables

**Figure 1 insects-12-01071-f001:**
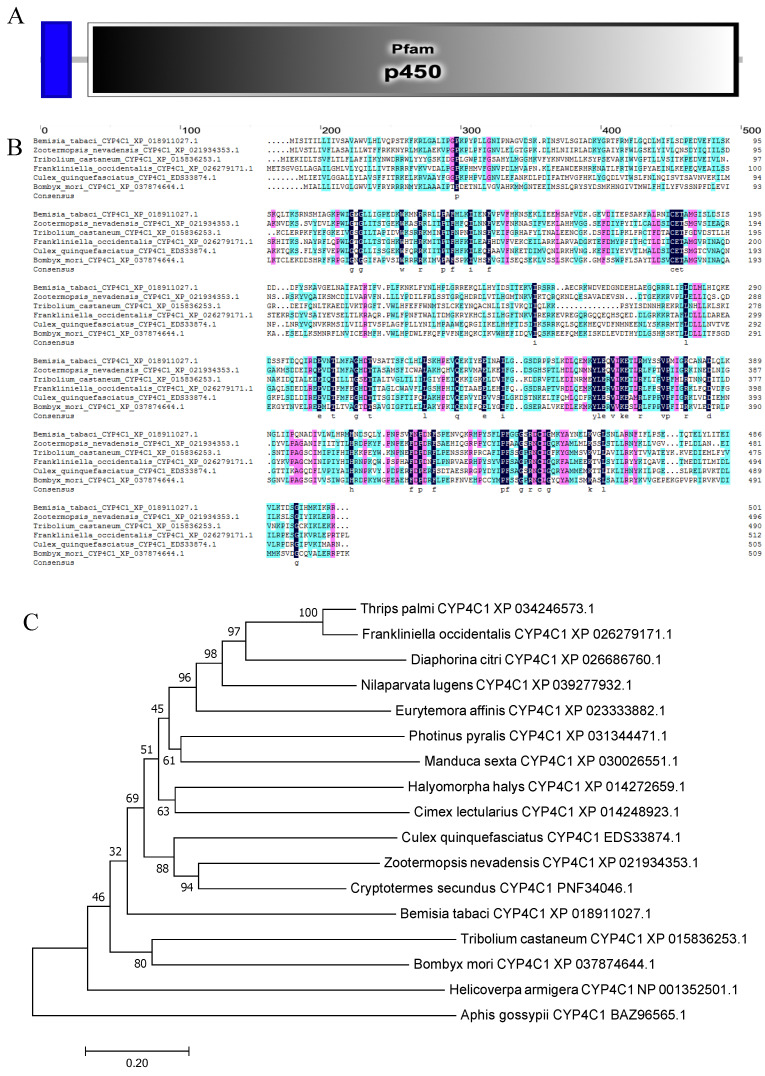
(**A**) The functional domain of *BtCYP 4C1*. (**B**) The multiple alignment of the deduced amino acids of *CYP 4C1* in *Bemisia tabaci*. (**C**) Phylogenetic analysis based on amino acid Scheme 4 *C1*.

**Figure 2 insects-12-01071-f002:**
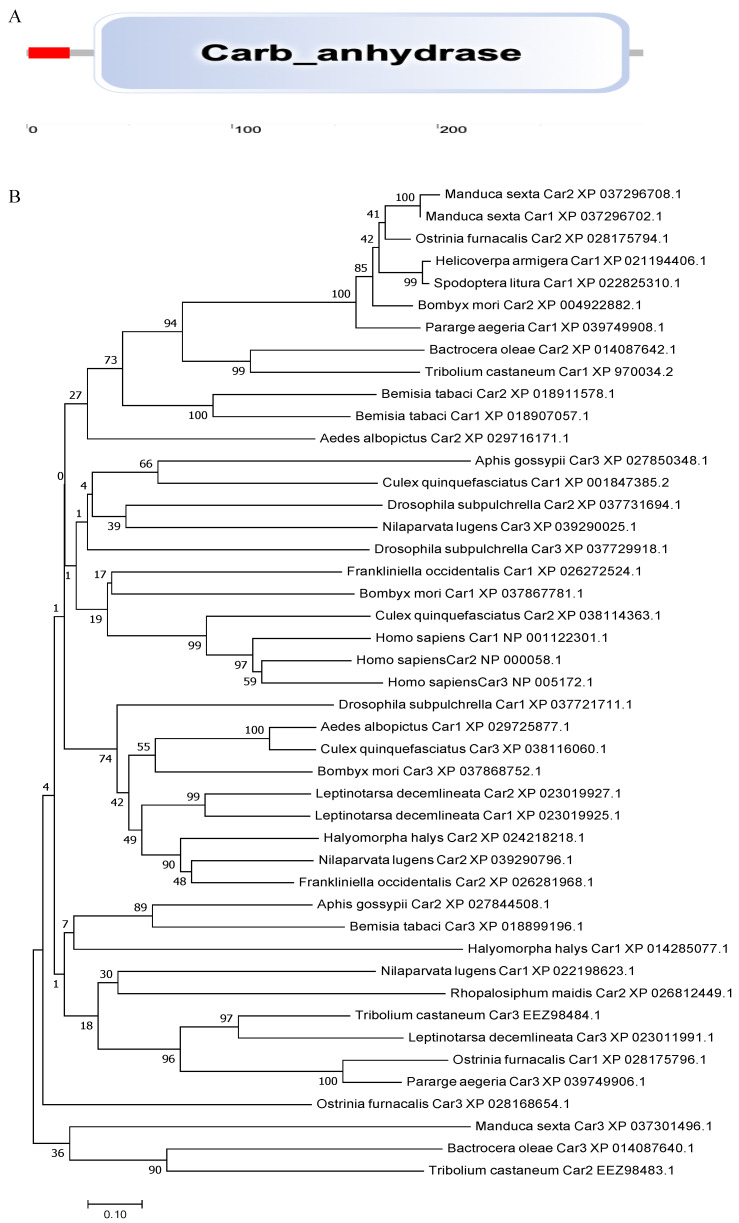
(**A**) The functional domain of *BtCYP 4C1*. (**B**) Phylogenetic analysis based on amino acid Scheme 1. Car2 and Car3. (**C**) Phylogenetic analysis based on amino acid sequences of Car1, Car2 and Car3 from Homo sapiens and *Bemisia tabaci*. (**D**) The multiple alignment of the deduced amino acids of Cars in *Homo sapiens* and *Bemisia tabaci*.

**Figure 3 insects-12-01071-f003:**
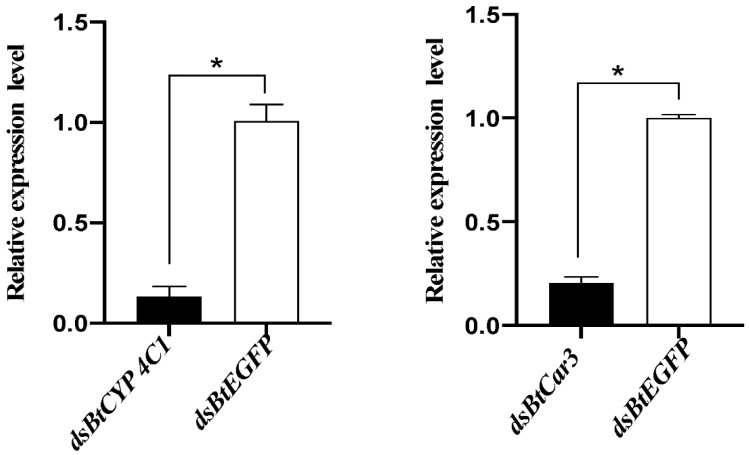
Effects of dsRNA treatments on mRNA expression in the *Bemisia tabaci*. Symbol * indicates significant differences at *p* < 0.05.

**Figure 4 insects-12-01071-f004:**
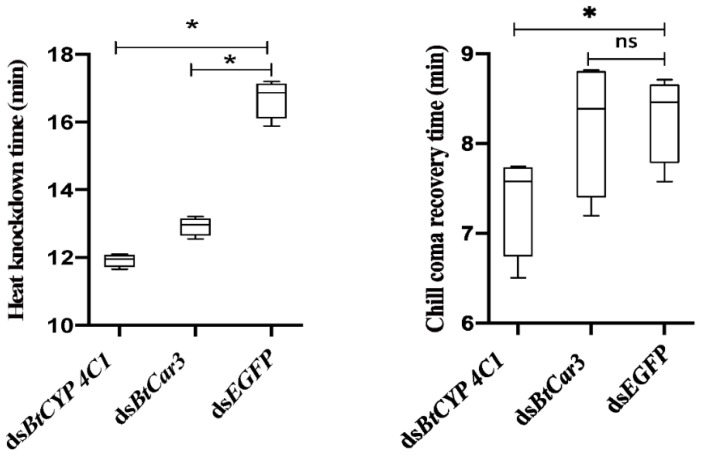
Trait means of heat and cold tolerances after feeding *Bemisia tabaci* with dsRNA of the *BtCYP 4C1* and *BtCar3* genes. Numbers are means ± 1 S.E. N = 200 in all cases. Symbol * indicates significant differences at *p* < 0.05.

**Table 1 insects-12-01071-t001:** Primer sequences of *BtCYP 4C1*, *BtCar3* and reference gene and the T7 primer sequences used to generate dsRNA in *Bemisia tabaci*.

Gene Name	Primer Name	Sequence Information (5′-3′)
*BtCYP 4C1*	*BtCYP 4C1*-F	GGAGGAAGTGACAGACC
	*BtCYP 4C1*-R	GTAAAAACGAATACGGG
*BtCar3*	*BtCar3*-F	GCGATGCGGTAGAGTTG
	*BtCar3*-R	CGAAAGGTTTGCTGAAGA
*EF1-α*	*EF1-α-F*	TAGCCTTGTGCCAATTTCCG
	*EF1-α-R*	CCTTCAGCATTACCGTCC
*BtCYP 4C1*	T7 + *BtCYP 4C1*-F	TAATACGACTCACTATAGGGTGGATAGGCGAGGGTCT
	T7 + *BtCYP 4C1*-R	TAATACGACTCACTATAGGGCGAATATTGCGTTGAGC
*BtCar3*	T7 + *BtCar3*-F	TAATACGACTCACTATAGGGGCCCCCTAAGGATTTATA
	T7 + *BtCar3*-R	TAATACGACTCACTATAGGGCTTCCGTTGTTGTCTGTCT

## Data Availability

The data presented in this study are available on request from the corresponding author.

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
