# Peer review of "The Role of Cytochrome P450 4C1 and Carbonic Anhydrase 3 in Response to Temperature Stress in Bemisia tabaci"

_insects, 2021, doi:10.3390/insects12121071_

Round 1
Reviewer 1 Report
The manuscript ‘The role of two key factors regulated by chromatin accessibility 2 in response to temperature stress in Bemisia tabaci’ investigated two genes, cytochrome P450 4C1 (BtCYP 4C1) and carbonic anhydrase 3 (BtCar3) which regulated by chromatin accessibility. In the study, the authors first obtained the full cDNA sequence of BtCYP 4C1 and BtCar3 and subsequently designed RNA interference (RNAi) experiment based on the mRNA sequence. They demonstrated the success of the RNAi that 80% reduction of expression of gene BtCYP 4C1 and BtCar3 with real-time qPCR. The author further demonstrated that the knock-down of BtCYP 4C1 gene by RNAi in Bemisia tabaci significantly reduced the heat tolerance (knockdown time at 45°C), but increase the cold tolerance (shorter recover time after chill-coma ( -5°C), while the knock-down of BtCar3 in Bemisia tabaci only affect heat tolerance, not the cold tolerance. The design and execution of the experiment were well performed and demonstrated the important roles of BtCYP 4C1 and BtCar3 in the tolerance of the extreme temperature of white fly. However, the major flaws of the manuscript are the 3.1 Differences in active chromatin associated with BtCYP4C1 and BtCar3 in B. tabaci under different temperatures and 3.2. Difference in transcription expression level of BtCYP4C1 and BtCar3 in B. These two sections were results from unpublished ATAC-seq experiment while the title, discussion and conclusion were drawn based these results. This is not acceptable for publication in the current form. I would suggest combining the full ATAC-seq experiment and current RNAi experiment as a full manuscript to justify the discussion and conclusion.
Following are some concerns of the manuscripts:
- Page 1 line 23-24 These results indicated that BtCYP 4C1 and BtCar3 are key regulators in the process of epigenetic regulation for temperature adaptation in B. tabaci. The current experiment cannot support the epigenetic regulation.
- Page 2 line 59-61: “this study found two highly significant genes from the ATAC-seq results, and verified their functions in temperature response.”. This study did not report the ATAC-seq full experiment, just referred the unpublished result.
- Page 2 line 62-65: “It is further confirmed that changes in chromatin openness can affect the temperature tolerance of whiteflies by regulating gene expression”. This study did not provide data evidence for the current study. This is refereeing the unpublished work.
- Page 2 line 92: please indicate if the qPCR was performed individually, or pooled samples. If using pooled samples, how many replicates were used for each treatment?
- Page 4 Results 3.1 Differences in active chromatin associated with BtCYP4C1 and BtCar3 in B. tabaci under different temperatures. This is not the result from this study.
- Page 4 Results: 3.2. Difference in transcription expression level of BtCYP4C1 and BtCar3 in B. tabaci under different temperatures. This is not the result from this study.
Page 10 line 262-24: “In this study, it is further confirmed that silencing cyto-chrome P450 4C1 genes affects the temperature tolerance of whitefly and the expression of CYP4C1 is regulated by chromatin accessibility.” The expression of CYP4C1 ie regulated by chromatin accessibility was not part of this study.
Reviewer 2 Report
The manuscript by Shen et al. provides data about the role of cytochrome P450 4C1 and carbonic anhydrase 3 in the process of epigenetic regulation for the B. tabaci tolerance in adverse temperature conditions.
In general, the manuscript is well written and organized. The findings contribute to the literature. Thus, I recommend the manuscript should be accepted for publication after some revision.
Below are a few comments that the authors should take into account in the revision of the manuscript.
Line 142. I don’t understand why the authors performed one-way ANOVA, and then used t-test to compare means. There are specific tests for post-hoc comparison of means after ANOVA returns significant differences (Tuckey, etc). Please explain your choice. In addition, did you examine the data to see whether ANOVA assumptions are met (e.g., normality, homogeneity of variance). If these assumptions are not met, the non-parametric methods should be used.
Figure 2. You should statistically compare the means of 21oC and 31oC.
Line 234. Add “(TRC)” after “recovery time”
Reviewer 3 Report
SUMMARY BY REVIEWER
Here, Shen et al. investigated the functional consequences of knocking-down two whitefly genes previously found within regulatory regions identified by a chromatin accessibility profiling method. The authors orally delivered dsRNAs at 300-500 ng/uL in a 10% sucrose solution to silence genes cytochrome P450 4C1 (CYP4C1) and carbonic anhydrase 3 (Car3). Following a 4-hour acquisition period, the authors selected 60% of whitefly cohort to measure knockdown by qPCR. The remaining 40% of the cohort was split evenly for an additional experiment consisting of the exposure of whiteflies to two temperature pressures at 45°C and -5°C, respectively. The authors found that knockdown was achieved at >80% for both genes compared to the GFP non-target control. The knockdown of both genes resulted in a reduced ability of whiteflies to withstand the 45°C regime. Negligible changes were observed regarding the low temperature regime between the dsRNA treatments and the GFP control.
WHY PUBLISH?
The research provides lines of evidence to support the role of two genes in the ability of Bemisia tabaci to withstand heat stress, which may explain the near-cosmopolitan distribution of the whitefly biotype studied here.
These findings together with findings that CYP4C1 might be involved in insecticide resistance (previously published) can have an impact of the management of whitefly invasive haplotypes.
This reviewer congratulates the author on the massive amount of work.
MAJOR CHANGES NEEDED BEFORE THE MANUSCRIPT IS PUBLISHABLE:
The title does not reflect the findings. The title should be something like: knockdown of cytochrome P450 4C1 and carbonic anhydrase 3 results if reduced heat stress tolerance in the whitefly Bemisia tabaci.
Several times throughout the text, the authors refer to previous unpublished data, to support the use of gene cytochrome P450 4C1 (CYP4C1) and carbonic anhydrase 3 (Car3) in the knockdown experiments. Such previous unpublished experiments were related to the chromatin accessibility profiling that led to the discovery of the genes silenced here. Such studied was vaguely mentioned in the methods, although longer bits were included in the results, with illegible figures. This reviewer suggests the authors to:
Option 1: remove all bits of text regarding the chromatin accessibility profiling and justify the use of genes in the introductions.
Option 2: fully include the chromatin accessibility profiling in all sections of the manuscript, describe the biological experiments, temperature regimes selection, sequencing method, assembly, filtering, annotation, and other that are relevant.
Provide evidence that reference gene EF1-a remained unchanged under these experimental conditions, and justify the use of one reference, which is against the MIQE guidelines for qPCR interpretation.
Provide evidence that the whiteflies used in these experiments were molecularly identified as the B biotype or MEAM species.
Error bars suggest no changes in Chill coma recovery time. However, the authors found differences between CYP4C1 and the GFP control.
OTHER MAJOR AND MINOR COMMENTS ARE INCLUDED BY SECTION
INTRODUCTION:
Lines 30-31: Even though debate continues on the mechanisms driving epigenetic processes. There are plenty of resources in the literature for the authors to concisely and scientifically re-write this sentence. Please see: Berger et al 2009, and Bird 2007.
Line 31: “chromatin accessibility”. I think the term needs to be introduced here, especially to the non-expert readers following Minnoye et al. 2021 as an example. I see lines 34-35 have a brief introduction but it is vague to me.
Line 33: I suggest that the authors please do not use “etc.” Either name all mechanisms or use words as “mainly” to name the most common. Also citations are required to support the claims.
Line 37-29: The authors would benefit from some English editing here. For instance: “Insects are ectothermic animals. Therefore, temperature AFFECTS THEIR is an important factor affecting all physiological activities of ectotherms, including their growth, development, reproduction and survival.” Use this example to fix throughout the text.
Line 39-40: The term “ecological environment” does not quite fit in this context. This reviewer suggests “field”, or “natural habitats”. A citation is needed.
Line 41-42: Please re-write this sentence to clarify that ectothermic attributes are not the only drivers of invasiveness in insect pests.
Line 44: Please delete the number 1 from MEAM1. The “species” MEAM2 no longer exists, therefore there is no need to include numbers next to the MEAM acronym. Also, some readers are not aware of/disagree with this nomenclature, please add “also referred to as the B biotype”.
Line 48: This was already said.
Line 51: Fix verbs tense throughout the text. Here it should be “ it was is found”
Line 53-54: Sounds imprecise to me. Also, a citation is required.
Line 54-56. Please cite your previous work. Also, it should be “chromatin opening in of B. tabaci” Also this is expected. Different environmental stimuli will lead to a different chromatin accessibility pattern with diverse regulatory regions. In fact, the chromatin accessibility profiling by ATAC-seq and DNase seq, are used to identify regulatory regions.
Lines 56-57: What is unknown are the regulatory regions remodeled as a response to temperature stress.
Line 57: Did the authors mean regulatory?
METHODS:
- tabaci is a cryptic species complex, therefore, identification is not possible using morphological characters. Were whiteflies identified using a DNA marker? It has been recently shown that MEAM or the B biotype comprises eight haplotypes, two of which can be found in China. See Paredes-Montero et al. 2021.
Line 72: MED or Q biotype is a cryptic species/mitochondrial type different to MEAM.
Line 75: “typically” implies several extraction replicates were carried out. Could the authors provide details on the biological and technical replicates carried out in these experiments?
Also, did the authors treated their RNA extractions with DNase before cDNA synthesis? This step is needed to avoid DNA contamination.
Lines 80-91: This section seems out of place. Unfortunately, there is a lot of detail missing here such as, how were sequences produced, assembled, filtered, etc. Alternatively, if this section belongs to a prior study, please cite the work.
Lines 103-104: Reference 24 does not support such claim. In fact, reference 24 was conducted on human organs, and the reference gene used in such study was GAPDH. More importantly, the MIQE guidelines (Benson et al 2009) for qPCR mention that normalization of expression using a single reference gene is not acceptable unless a study has been conducted that yields clear evidence that EF1-a does not vary under the experimental conditions used in this work. Could the authors please provide evidence that EF1-a remained unchanged under different temperature-stress conditions, e.g., qPCR ct curves, or genorm, normfinder and bestkeeper analysis of ct values?
It is unclear how genes BtCYP4C1 and BtCar3 were selected. They first time the acronyms showed up was in table 1. Please introduce them briefly in the last paragraph of your introduction.
Line 115: I think synthesis is a better verb than generate in this context.
Lines 115-116: Oh, here I see the genes of interest for the first time! But no acronyms. Could the authors please standardize the way these genes are mentioned throughout the text?
Line 121: Did the authors tested different dsRNA concentrations? Please add details on the titrations used before optimal concentrations were determined. Also, Tseng’s reference is not in the references list, 10% is a little too low for a whitefly artificial diet.
Line 126: a translation issue here, whiteflies are not flies, please remove.
Lines 127: How many whiteflies per glass tube?
Line 129: move toward …where? Maybe the direction of light?
Line 130: It was mentioned above that RNA was extracted from 200 whiteflies. Now, 180 were processed for qPCR.?
I am concerned that 4 hours is too short of an acquisition time unless a starvation period was given prior exposure of whiteflies to the dsRNA diet. Otherwise, how can the authors make sure that all whiteflies fed on the diet?
Line 131: Please avoid using RT-qPCR, qPCR is enough.
300 individuals ≠ 180 for qPCR + 40 high temperature + 40 low temperature. I guess the remaining 40 whiteflies died. Please clarify if that is so. Also clarify that 60 whiteflies were used per treatment to measure knockdown.
Line 132: Do not use the term flies to refer to whiteflies. Flies are insects that belong to the order Diptera.
Line 132: For how long did the authors exposed the flies to the temperature regimes? And clarify whether whiteflies were immediately processed to measure knockdown.
Line 134: So, did the authors use 300 x 3 treatments and control + 6 biological replicates = 5400 whiteflies.?
Line 135: It is crucial that the authors cite these preliminary studies.
Lines 138-139: Please clarify what analyses were done in GraphPad Prism, and which ones in Microsoft Excel.
Line 149: Unpublished? However, submitted for review to become published. I recommend the authors to add a detailed section in the methods to describe these experiments otherwise, if it has been published section 3.1 must be removed from this manuscript, and information about these genes must be included in the introduction.
Section 3.2 in the results is really what this manuscript is about. Please change the title to reflect the specific findings in this manuscript.
Line 160: It is very unfortunate, but the authors cannot base their findings on research that has not been published and that has not been submitted for review. As a reviewer, I must look at the whole data to evaluate this manuscript objectively and critically. The sections pertaining to Figure 1 must be deleted from this manuscript. Also, delete section 2.3, which I am guessing refers to the work you did previously. Either combine your previous work and the knockdown data in a single manuscript or delete all the sections regarding chromatin remodeling from this manuscript.
Line 226. There was just one control group.
Line 228. This reviewer is impressed by the knockdown achieved after exposing whiteflies to dsRNAs for only 4 hours. Could the authors compare these results to similar ones available in the literature.?
Line 229: Please, be specific on the knockdown values for each gene.
Line 244: Not Bemesia, Bemisia. Also, please italicize Bemisia tabaci throughout the text.
Line 244: Figure 6, right panel. Based on the standard error bars, there are not statistical differences among your treatments and control. Please provide evidence of the specific statistical analyses that supported differences. Also, where does 200 come from? You used 40 whiteflies per treatment per biological replicate, and you carried out the experiment 6 times, that adds up to 240 whiteflies per replicate used for temperature stress treatments. Anyhow, the total number of whiteflies used is irrelevant herein. The authors must clarify on the biological and technical replicates used.
Line 248: This is erroneous, the B biotype is native in the middle east.
Lines 249: mutations seem out of place, not need in this context.
Line 251: non-negligible role? I don’t think is needed.
Line 254: in instead of from.
Lines 254-261. Nematode and Rose examples are irrelevant in the context of this manuscript. Please add more examples regarding organism more closely related to whiteflies such as aphids, psyllids, or other insects. Here, the authors discussed that previous works on different organisms showed that knockdown of CYP4C1 orthologs resulted in decreased survival rates. Here you kind of showed the opposite. Knockdown of gene CYP4C1 in whiteflies, at 80%, resulted in whiteflies recovering faster from freezing coma than the control. Therefore, under such scenario, a high expression of CYP4C1 negatively affects the whitefly to withstand low temperatures.
In fact, your charts do not show any statistical differences regarding the “chill coma recovery time”, meaning that knockdown of CYP4C1 and Car3 did not have an effect on the ability of whiteflies to withstand low temperatures. This could make perfect sense, because the genes CYP4C1 and Car3 were selected from chromatin accessibility patterns subjected to three temperature regimes, none of them below 21°C, yet here the authors tested -5°C. This reviewer does see a clear effect on heat stress tolerance, which is significantly increased after knockdown of genes CYP4C1 and Car3 at >80%.
Lines 273-276: The authors must decide whether Shen’s unpublished data will be combined with knockdown data in a single paper. I suggest the authors to incorporate the chromatin accessibility profiling in the methods section, including details on the experiments carried out to collect the whiteflies for ATAC-seq at different temperature regimes, explain the rational on the selection of temperatures for the chromatic accessibility profiling, introduce to the sequencing methods, and improve the sequences analysis section.
Line 274: it was found.
Round 2
Reviewer 1 Report
I am satisfied with the revised version that has addressed all my concerns for the manuscript and agreed for publication.
Author Response
Once again, thank you very much for your contribution to this manuscript.Reviewer 3 Report
GENERAL ASSESSMENT BY REVIEWER
This reviewer considers the authors have done a great job in incorporating changes suggested during the first round of revisions. This reviewer considers the article is in better shape than its previous version and therefore endorses publication after the points below are considered.
OTHER CHANGES NEEDED
Please spell out gene names in the title, BtCYP and BtCar are not commonly used acronyms.
Paragraph 1 no longer belongs in this manuscript. Please remove.
Line 58-59: not needed. Lines 57- 63 should be: “ Several studies showed that temperature adaptability has likely allowed invasive B. tabaci haplotypes to expand to new environmental niches. These adaptation processes have been described as transient phenotypic changes, likely mediated by epigenetic regulation, wherein the genetic sequence remains unchanged.”
Line 65-67: Should be “These results support the hypothesis that the adaptability of B. tabaci to temperature stresses is mediated epigenetically. In such a specific chromatin accessibility pattern, regulatory regions were found to harbor thirty genes of interest among which, cytochrome P450 4C1 (BtCYP 4C1) and carbonic anhydrase 3 (BtCar3) we selected based on …..” Please add rationale on which you based to select these two genes out of 30. Provide a brief introduction on the functional role of these genes in Bemisia tabaci or other insects.
Line 69-70: The last sentence is wrong. I believe the authors meant to write that the predicted role of these genes in the response of B. tabaci to temperature stress has not been confirmed by functional studies.
Line 71-72: Again, please change “regulatory factors” throughout the text. Use the gene names specifically: Example: In order to verify the role of BtCYP4C1 and BtCar3 in the response of B. tabaci to temperature stresses, silencing of these genes was carried out using an RNAi approach. Whiteflies B. tabaci were fed on specific dsRNAs followed by exposure to heat and cold stress regimes. The timing of recovery from temperature shocks was recorded as a measurement of the effect of the dsRNAs on whiteflies.”
Line 133: To synthesize (I am sorry I suggested synthesis before)
Line 134: How long were dsRNAs for BtCYP and Car3?
Line 140: It is extremely important to know what concentration of dsRNAs did you use to knockdown BtCYP and what concentration for Car3. A range at 0.3-0.5 ug/uL is unacceptable if one wants to replicate these experiments.
Line 249-250: I must insist that the length, and concentration of dsRNAs used PER GENE, as well as the resulting knockdown, must be clarified for each gene. A range is not acceptable. I understand both genes resulted in knockdown above 80%, but was BtCYP 82% and Car3 95%? Please clarify.
Line 262: change verbs to past tense.
Lines 254-268: Please revise these sentences. The authors provided the picture below of their statistical test, where they showed degrees of freedom (df) is 6. However, degrees of freedom (df) equal N-1, where N is the number of replicates. In this study 6 replicates were carried out. Therefore, degrees of freedom is 5. A “T” value of 2.482 with degrees of freedom 5 does not support differences at the 95% confidence level. Higher cold resistance than the wild type is not supported.
Unfortunately, a good reference gene in experimental setting “A” may not work well in experimental setting “B”. Therefore, reference genes must be assessed in advance to find the most optimal ones. If the authors do not provide strong evidence of the low variability of EF1-a in their specific setting, the knockdown results are not reliable. Also, reference 25 on the most optimal reference genes, showed EF1-a worked fine in the MED (Q biotype) species, however, the authors worked with a different species, the MEAM (B biotype). This reviewer will not recommend rejection based on this aspect. However, the authors must clarify that knockdown may change if a different reference gene is used to normalize gene expression.
Replace “tabacis” with tabaci. Also, replace “with…” with “using…” (line 86)
